# Enhancing Heat Dissipation of Photoluminescent Composite in White-Light-Emitting Diodes by 3D-Interconnected Thermal Conducting Pathways

**DOI:** 10.3390/mi13081222

**Published:** 2022-07-30

**Authors:** Puzhen Xia, Bin Xie, Xiaobing Luo

**Affiliations:** 1School of Energy and Power Engineering, Huazhong University of Science and Technology, Wuhan 430074, China; pzxia@hust.edu.cn; 2School of Mechanical Science and Engineering, Huazhong University of Science and Technology, Wuhan 430074, China

**Keywords:** white-light-emitting diodes, photoluminescent composite, heat dissipation, hexagonal boron nitride, 3D network

## Abstract

The photoluminescent composite, which consists of micro-/nanoscale photoluminescent particles and a polymer matrix, plays a key role in optical wavelength conversion in white-light-emitting diodes (WLEDs). Heat is inevitably generated within the composite due to the energy lost through conversion and cannot be easily dissipated due to the extremely low thermal conductivity of the polymer matrix. Consequently, the composite suffers from a high working temperature, which severely deteriorates its optical performance as well as its long-term stability in WLEDs. To tackle this thermal issue, in this work three-dimensional (3D)-interconnected thermal conducting pathways composed of hexagonal boron nitride (hBN) platelets were constructed inside a photoluminescent composite, using a simplified bubbles-templating method. The thermal conductivity of the composite was efficiently enhanced from 0.158 to 0.318 W/(m∙K) under an ultralow hBN loading condition of 2.67 wt%. As a result, the working temperature of the photoluminescent composite in WLEDs was significantly reduced by 32.9 °C (from 102.3 °C to 69.4 °C, under 500 mA). Therefore, the proposed strategy can improve the heat accumulation issue in photoluminescent composites and thus improve the optical stability of WLEDs.

## 1. Introduction

White-light-emitting diodes (WLEDs) have been extensively adopted for general lighting, backlight displays, visible light communication, medical lighting, etc., due to their outstanding characteristics such as superior color quality, long life span, and high energy efficiency [1,2,3]. Figure 1 shows the typical packaging structure of a WLED module, which mainly consists of a blue gallium nitride (GaN) LED chip, a photoluminescent composite, a lead frame, a heat sink, and encapsulation. After being powered on, the LED chip emits blue light first. Subsequently, some of blue light photons that hit the photoluminescent particles are absorbed by the particles in the photoluminescent composite, then converted into yellow and red light. The optical wavelength of the converted light is determined by the bandgap of the photoluminescent particles. Moreover, there are some blue light photons that transmit through the composite without hitting the photoluminescent particles. Eventually, the combination of transmitted blue light and converted yellow/red light generates white light.

Alongside the aforementioned white light generation process, the LED chip generates heat due to the occurrence of different kinds of non-radiative recombination and other causes of photon annihilation [4]. Photoluminescent particles including phosphors, microspheres, and quantum dot (QD) nanoparticles inevitably generate heat due to Stokes loss, non-radiative recombination loss, and scattering loss [5]. Since these particles are surrounded by a polymer matrix whose thermal conductivity is extremely low, their heat cannot dissipate efficiently. As a result, the working temperature of the photoluminescent particles can reach 130 °C, which exceeds the junction temperature of an LED chip [6,7]. This high temperature will cause photoluminescence degradation or even the quenching of phosphors/QDs, thus severely deteriorating the optical performance as well as the long-term stability of the WLED. To solve this thermal issue, the heat dissipation process inside the photoluminescent composite needs to be enhanced.

However, the conventional thermal management solutions for WLEDs such as thermal interface materials (TIMs) [8], liquid cooling [9], thermoelectric cooling (TEC) [10], and structural optimization [11,12], etc., mainly focus on the peripheral heat dissipation process from the chip substrate to the ambient environment. Therefore, these strategies are unable to enhance the heat dissipation process that occurs inside the photoluminescent composite. Recently, highly thermally conducting hexagonal boron nitride (hBN) platelets were incorporated into photoluminescent composites to reinforce the heat dissipation process without sacrificing the optical efficiency of the WLEDs [13,14]. These hBN platelets are effective in enhancing the thermal conductivity of photoluminescent composites. For example, a hBN-filling load of 4.3 wt% can enhance the thermal conductivity by 58% [13]. Though effective, the filling fraction of hBN should be maintained below a relatively low level (less than 8 wt%), otherwise the optical performance of the WLEDs will decrease due to the scattering effect of the hBN [13]. According to the percolation theory [15], such a low filling load cannot form continuous thermal conduction pathways within the composite; thus, the promotion of heat dissipation is limited.

To tackle the aforementioned issue, in this work we proposed a simplified bubbles-templating method to construct three-dimensional (3D)-interconnected thermal conducting pathways inside the photoluminescent composite under an extremely low hBN filling load of 2.67 wt%. Owing to the existence of cellulose nanofibers (CNFs) with long molecular chains that support and stabilize the 3D microstructures, the fabricated 3D-interconnected pathways show robust mechanical stability during the whole process. The thermal conductivity of the composite was efficiently enhanced by 100% from 0.158 to 0.318 W/(m∙K). Consequently, the working temperature of photoluminescent composites in WLEDs was significantly reduced by 32.9 °C (from 102.3 °C to 69.4 °C, under 500 mA), without sacrificing the optical performance of the WLEDs. So far, rarely has a similar strategy been carried out to realize remarkable temperature reductions in WLEDs while maintaining their optical performance. The proposed thermal management strategy for photoluminescent composites can relieve the heat accumulation issue inside the WLEDs and thus push forward the development of high-power WLEDs.

## 2. Materials and Methods

### 2.1. Materials

hBN platelets with an average size of 45 μm and thickness of 1 μm were purchased from Momentive (Shanghai, China). Alkyl Polyglucoside (APG) was purchased from Sigma-Aldrich (49122, Shanghai, China) to be used as the foaming agent. Cellulose nanofiber (CNF, 1 wt%) was purchased from Tianjin Wood Spirit Biotechnology Co., Ltd. (Tianjin, China), to be used as the stabilizing agent. Five-watt LED chips with peak wavelengths of 465 nm were purchased from TYAOLED Co., Ltd (Shenzhen, China). Phosphors with an average diameter of 13 μm and peak wavelength of 538 nm were purchased from Intermatix (Fremont, CA, USA). Red-emissive QDs with an average diameter of 6 nm and peak wavelength of 626 nm were purchased from Guangdong Poly Optoelectronics Co., Ltd (Jiangmen China). Bicomponent silicone was purchased from Dow Corning (SYLGARD 184, A:B = 10:1, Midland, MI, USA). All of the materials were used without modifications.

### 2.2. Fabrication 3D-Interconnected Thermal Pathways and WLEDs

Figure 2 shows the schematical fabrication process used for the 3D-interconnected thermal pathways. First, a 100 mL beaker was placed onto the magnetic stirrer and 20 mL CNF solution was added, with the stirring speed controlled at 300 rpm to ensure the raw materials were uniformly mixed. Then, for the foaming process, 0.5 mL APG, 2 g hBN platelets, and 1 g phosphor powder were added into the solution and the stirring speed was increased to 1500 rpm for 10 min. During this foaming process, massive air bubbles were generated and uniformly distributed into the mixture due to the foaming function of APG, which resulted in the volume expansion of the mixture. In the meantime, the hBN platelets were extruded by the bubbles and thus the hBN made contact to form a 3D-interconnected network. Owing to the stabilization of CNF, the mixture gradually transformed into a hydrogel, in which the hBN, phosphor, and bubbles were uniformly dispersed. After foaming, the hydrogel was subjected to a freeze-drying process under −55 °C and 20 Pa for 24 h. As a result, the hydrogel transformed into aerogel with massive 3D-interconnected hBN networks and 3D-interconnected voids. Subsequently, to fabricate the 3D/hBN-network-based photoluminescent composite (3D/hBN-composite), QDs-silicone gel was immersed into the aerogel under the assistance of a vacuum to fill the air voids. Finally, the mixture was cured by heating it to 100 °C for 30 min. To fabricate the 3D/hBN-composite-based WLEDs (3D/hBN-WLEDs), the hydrogel was directly poured onto the cavity of the blue LED, and the other steps were followed as above. However, in order to compare the influence of different hBN distributions on the optical and thermal performances of WLEDs, traditional WLEDs (T-WLEDs) without hBN and WLEDs with an identical weight fraction but randomly distributed hBN (R/hBN-WLEDs) were also fabricated. It should be noted that CNF was not included in the fabrication of T-WLEDs and R/hBN-WLEDs since the foaming process is not necessary for the fabrication of the T-composite and R/hBN-composite.

### 2.3. Thermal Simulation of Photoluminescent Composites and WLEDs

To analyze the influence of different hBN distributions (without hBN, randomly distributed hBN, and a 3D/hBN network) on the reinforcement mechanisms of the heat transfer process within these composites, thermal models of the photoluminescent composite without hBN (T-composite), the composite with randomly distributed hBN (R/hBN-composite), and the 3D/hBN-composite were established, as shown in Figure 3a–c. The physical size of these models was 700 μm × 350 μm, and the left and right boundaries of the models acted as the thermal insulation. The bottom temperature of the models was fixed at 120 °C, and the heat-transfer coefficient along the top boundary was set at 200 W/(m^2^·K). The ambient temperature was fixed at 20 °C. The diameter and thickness of the hBN platelets were set at 45 μm and 1 μm, respectively, according to their actual size. The hBN filling fraction was the measured value (2.67 wt%, in this study). The thermal conductivity of hBN and silicone gel were 300 W/(m∙K) and 0.16 W/(m∙K), respectively.

In addition, to study the effect of these different hBN configurations on the thermal performance of WLEDs, thermal models of T-WLEDs, R/hBN-WLEDs, and 3D/hBN-WLEDs were also established, as shown in Figure 3d. These WLEDs were bonded onto aluminum alloy fins for heat dissipation. To reduce the simulation consumption, only half of the models were used in the simulations, and the symmetry plane acted as the thermal insulation boundary. An air domain whose size is three times larger than the model was set to surround the model to simulate the natural thermal convection process. The ambient temperature was fixed at 20 °C. There were two kinds of heat sources in the model: one was the LED chips, and the other was the photoluminescent composites. The heat power of these heat sources was calculated based on the optical energy loss of the WLEDs, which was measured by an integrating sphere [13].

### 2.4. Characterization and Measurements

The microscopic morphologies of the aerogel and composites were characterized using a scanning electron microscope (SEM, Nova NanoSEM 450, FEI, Maryland, USA). The 3D microstructures of the aerogel were characterized using an X-ray microscope (XRM) (skyscan 1172, Bruker, Massachusetts, USA). The thermal diffusivity (α), specific heat (Cp), and density (ρ) of the composites were measured using a laser flash device (LFA-457, Netzsch, Bavaria, Germany), a differential scanning calorimeter (DSC, Diamond DSC, Perkin Elmer, Waltham, MA, USA), and an electronic densitometer, respectively. The thermal conductivity (λ) of the composites was simultaneously determined by the thermal diffusivity (indicates the speed of heat transport), specific heat, and the density, and can be calculated by the equation, λ = α∙Cp∙ρ. The surface temperature distributions of the WLEDs were measured using an infrared thermal imager (SC-620, FLIR, Shanghai, China). The optical performances of the WLEDs were measured using an integrating sphere (ATA-1000, Everfine Co., Ltd., Hangzhou, China).

## 3. Results and Discussion

### 3.1. Morphological Analysis of the 3D-Interconnected hBN Network

During the construction of the 3D-interconnected hBN network, the generation and stabilization of air bubbles was critical. At the beginning of the foaming process, the volume fraction of the air bubbles was insufficient, and the hBN platelets were mainly randomly distributed in the solution, making it unlikely for them to connect with each other to form an interconnected network. With the increasing amount and volume fraction of air bubbles, each air bubble occupies part of the space and pushes aside the hBN platelets nearby. This effect makes the hBN platelets more likely to be interconnected to form a 3D thermal conducting network. However, if the volume fraction of the air bubbles is superfluous, the excessive air bubbles may break the hBN network, thus interrupting the continuous thermal conduction process. Therefore, the volume fraction of the air bubbles in the hydrogel should be carefully controlled. After several attempts and adjustments, the volume fraction of the air bubbles was fixed at 75%, and the relative deviation in the volume fraction of the air bubbles in different batches was controlled at less than 5%. Figure 4 shows the images of the prepared 3D/hBN hydrogel, aerogel, and network.

As shown in Figure 4a,b, the original volume of the mixture was 20 mL. After foaming, the mixture expanded to 80 mL, indicating that the volume of air bubbles was 60 mL. Figure 4c shows the optical microscopic photograph of the hydrogel, where the air bubbles were uniformly dispersed within the hydrogel. Figure 4d,e show the XRM image and SEM image of the prepared 3D/hBN network aerogel, respectively. It can be seen that after freeze-drying, the 3D/hBN aerogel has retained the original shape of 3D/hBN hydrogel. This is mainly attributed to the crosslinking function of CNF which provides mechanical strength to the aerogel. It can be seen from the SEM image that the size of the air voids in the aerogel ranges from 100 to 400 μm, and the voids are interconnected with each other. Besides the voids, the skeleton consisting of hBN and CNF is also 3D-interconnected, which may facilitate the dissipation of heat generated from the photoluminescent particles. Figure 4f shows the SEM images of the 3D/hBN-composite, in which the previous air voids in the 3D/hBN aerogel were completely filled by the QDs-silicone. From the enlarged SEM image in Figure 4f, it can clearly be seen that the hBN platelets have formed several continuous channels to enable efficient heat dissipation. However, from the SEM images of the T-composite (Figure 4g) and the R/hBN-composite (Figure 4h), only randomly distributed phosphor particles and/or hBN platelets are observed, which also indicates the successful formation of a 3D hBN network using the air-bubbles-templating method.

### 3.2. Optical and Thermal Performance Analysis

Figure 5 shows the thermal conductivity of the T-composite, R/hBN-composite, and 3D/hBN-composite under different ambient temperatures. It can be seen that at an ambient temperature of 25 °C, the thermal conductivity of the T-composite is only 0.16 W/(m∙K). With the addition of the 2.67 wt% randomly distributed hBN platelets, the R/hBN-composite features an elevated thermal conductivity of 0.185 W/(m∙K). With the introduction of the air bubbles, the thermal conductivity of the 3D/hBN-composite reached 0.316 W/(m∙K). Therefore, when compared with the T-composite and R/hBN-composite, the thermal conductivity of the 3D/hBN-composite realized an enhancement of 100% and 71%, respectively. Figure 6a–c show the simulated temperature distribution and heat flux distribution of the T-composite, R/hBN-composite, and 3D/hBN-composite, respectively. In Figure 6b, the hBN platelets are randomly dispersed in the polymer matrix, thereby the hBN cannot form any continuous chains. As a result, the heat within the composite cannot be effectively dissipated. Whereas, in Figure 6c, the hBN platelets are in contact with each other and form an interconnected network owing to the extrusion of air bubbles. As a result, the heat flux density in the network is an order of magnitude higher than that in the matrix, which indicates that the constructed 3D/hBN network is effective in reinforcing the heat dissipation of photoluminescent composite.

To study the effect of different hBN configurations on the optical and thermal performances of WLEDs, the T-WLEDs, R/hBN-WLEDs, and 3D/hBN-WLEDs were fabricated. The phosphor weight fraction in these WLEDs was kept to 10 wt%, and the hBN weight fraction in R/hBN-WLEDs and 3D/hBN-WLEDs was kept to 2.67 wt%. The correlated color temperature (CCT) of these WLEDs was adjusted to around 5500 K (neutral white light), by slightly tuning the usage of the QDs.

Figure 7 shows the optical spectra and key performance data of these WLED samples under 500 mA illumination. It can be seen that under the similar CCT, the luminous efficacy (LE) of these WLEDs is maintained at around 85 lm/W, which indicates that 2.67 wt% of hBN will not influence the optical efficiency of WLEDs. Among these samples, the 3D/hBN-WLEDs demonstrate a higher LE than the R/hBN-WLEDs. This can mainly be attributed to the massive 3D-interconnected voids (that were filled by QDs-silicone gel afterwards) in the 3D/hBN-composite, which enabled the unhindered transmission of light rays and reduced the optical energy loss. It was also found that, when the filling fraction of hBN increased to 4.5 wt%, the LE of the 3D/hBN-WLEDs decreased to 75 lm/W. This was mainly because the further addition of hBN introduces an extra scattering effect and thus increases the optical energy loss within the composite. Therefore, to guarantee the optical performance of WLEDs, the incorporated weight fractions of hBN should also be controlled at a low level.

Figure 8a displays the simulated temperature distributions of these WLED samples under 500 mA illumination. It can be seen from the distributions that the highest temperature in all these WLEDs is located in the topset area of the WLED. This temperature distribution has confirmed that the heat generated by photoluminescent particles is accumulated within the composite due to its low thermal conductivity, thus resulting in a local hotspot. By comparing the temperature distributions of T-WLEDs and R/hBN-WLEDs, it can be concluded that the hotspot temperature in the photoluminescent composite reduced from 105.6 °C to 91.2 °C after the incorporation of 2.67 wt% randomly dispersed hBN platelets. Furthermore, with the incorporation of a 2.67 wt% 3D/hBN network, the hotspot temperature was significantly reduced to 74.5 °C, which proved that the 3D-interconnected structure is more effective than the random structure in reinforcing the heat dissipation within the photoluminescent composite.

To validate the above simulation results, the steady-state surface temperature fields of these WLEDs were measured using an infrared thermal imager. During the measurements, the surface emissivity of the photoluminescent composite was to 0.96, according to our previously described calibration test [16]. The distance between the IR camera and the samples was set to 0.3 m. After 3 min of illumination, the surface temperature of these samples reached a steady state (the hotspot temperature variation was less than 1% within 1 min). Figure 8b displays the measured temperature distributions of these WLEDs under 500 mA illumination. It can be seen that the relative hotspot temperature error between the measurements and simulations is less than 10%, which indicates that the measured results have confirmed the aforementioned simulation results. Judging from the experimental results, the highest temperature in the photoluminescent composite was reduced by 32.9 °C under 500 mA, benefitting from the existence of a 3D/hBN network. Therefore, both the simulations and experiments have shown the effectiveness of a 3D/hBN network for the heat dissipation reinforcement of photoluminescent composites in WLEDs.

## 4. Conclusions

Photoluminescent composites are the key component in WLEDs. They function as the color conversion layer which converts blue light into yellow/red light, as well as the color mixing layer which generates white light. However, the heat generated by the photoluminescent particles is not easily dissipated outside the composite because of the extremely low thermal conductivity of the polymer matrix. This phenomenon results in a high working temperature in the composite, and thus severely deteriorates its optical performance as well as the long-term stability of the WLEDs. To tackle this thermal issue, 3D-interconnected thermal conducting pathways composed of hBN platelets were constructed inside the photoluminescent composite, using a simplified bubbles-templating method. Benefiting from the extrusion of air bubbles, massive 3D-interconnected hBN pathways were constructed under an ultralow BN loading of 2.67 wt%. Consequently, the thermal conductivity of the composite was efficiently enhanced from 0.158 to 0.318 W/(m∙K). As a result, the working temperature of the photoluminescent composite in WLED was significantly reduced by 32.9 °C (from 102.3 °C to 69.4 °C, under 500 mA). Therefore, the proposed strategy can relieve the heat accumulation issue in the photoluminescent composite, and thus improve the optical stability of WLEDs.

## Figures and Tables

**Figure 1 micromachines-13-01222-f001:**
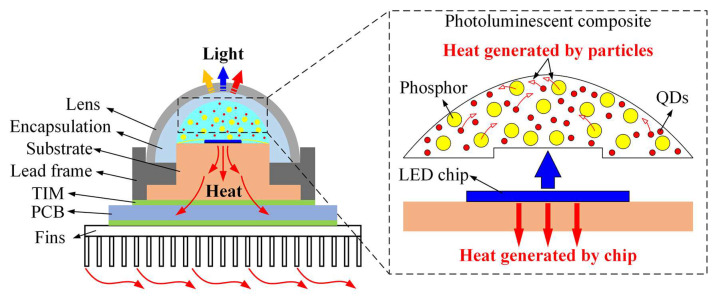
Schematic showing the packaging structure and heat generation mechanisms of WLEDs.

**Figure 2 micromachines-13-01222-f002:**
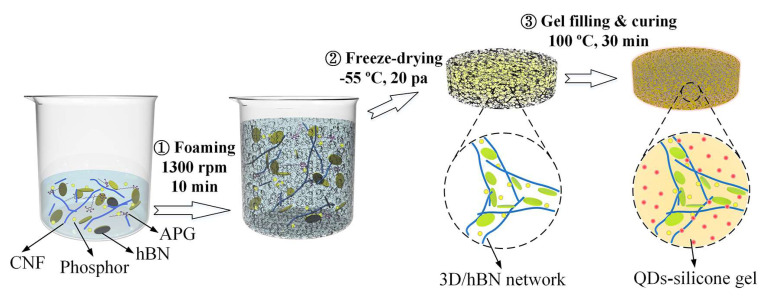
Schematic showing the fabrication process of 3D/hBN network and 3D/hBN-composite.

**Figure 3 micromachines-13-01222-f003:**
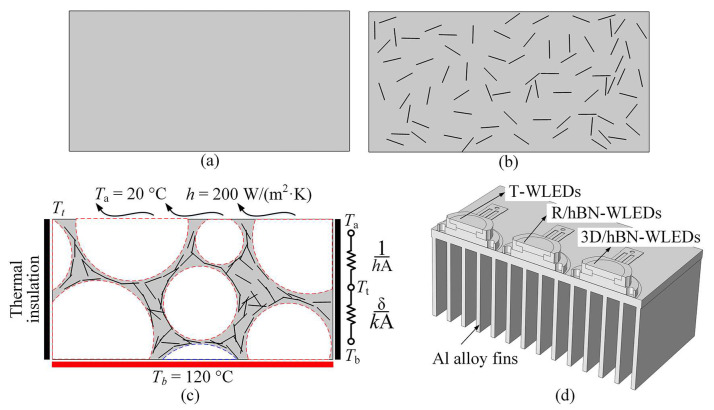
(**a**–**c**) Schematics showing the physical model and boundary conditions of T-composite, R/hBN-composite, and 3D/hBN-composite, respectively. (**d**) Schematic showing the thermal simulation models of these WLEDs.

**Figure 4 micromachines-13-01222-f004:**
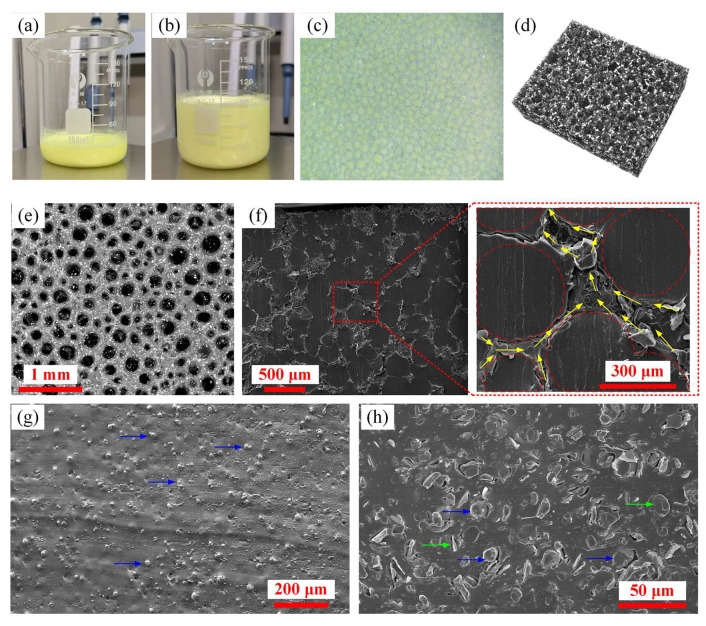
(**a**,**b**) Photographs of the hydrogel before and after foaming, respectively. (**c**) Optical microscopic photograph of the hydrogel after foaming. (**d**) XRM image of the 3D/hBN aerogel. (**e**) SEM image of the 3D/hBN aerogel. (**f**) SEM images of the 3D/hBN-composite. The red dashed circles indicate the locations of previously existing air voids before filling with QDs-silicone, and the yellow arrows indicate the heat transfer pathways established by hBN platelets. (**g**) SEM image of the T-composite. (**h**) SEM image of the R/hBN-composite. The blue and green arrows in (**g**,**h**) indicate the locations of phosphor particles and hBN platelets, respectively.

**Figure 5 micromachines-13-01222-f005:**
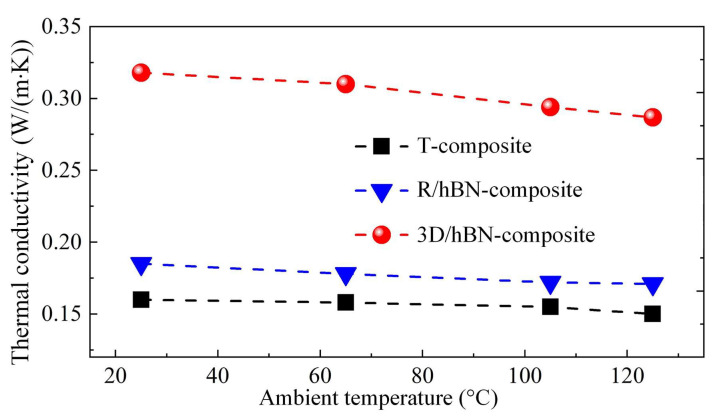
Measured thermal conductivities of the T-composite, R/hBN-composite, and 3D/hBN-composite under different ambient temperatures.

**Figure 6 micromachines-13-01222-f006:**
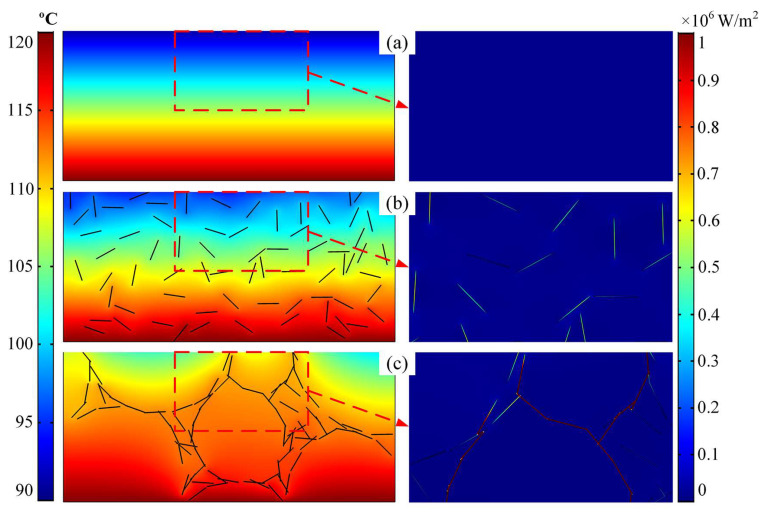
(**a**–**c**) Simulated temperature distribution (**left**) and heat flux distribution (**right**) of the T-composite, R/hBN-composite, and 3D/hBN-composite, respectively.

**Figure 7 micromachines-13-01222-f007:**
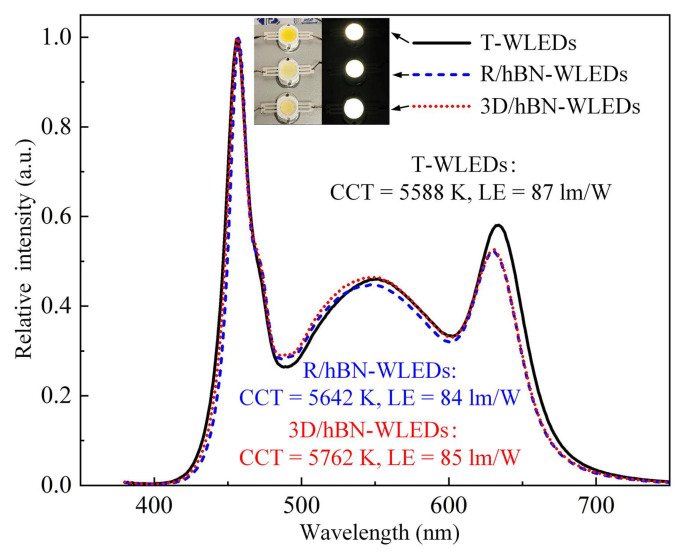
Optical spectra and key performance data of these WLEDs samples under 500 mA illumination. Insets show the photographs of these WLEDs working and not working.

**Figure 8 micromachines-13-01222-f008:**
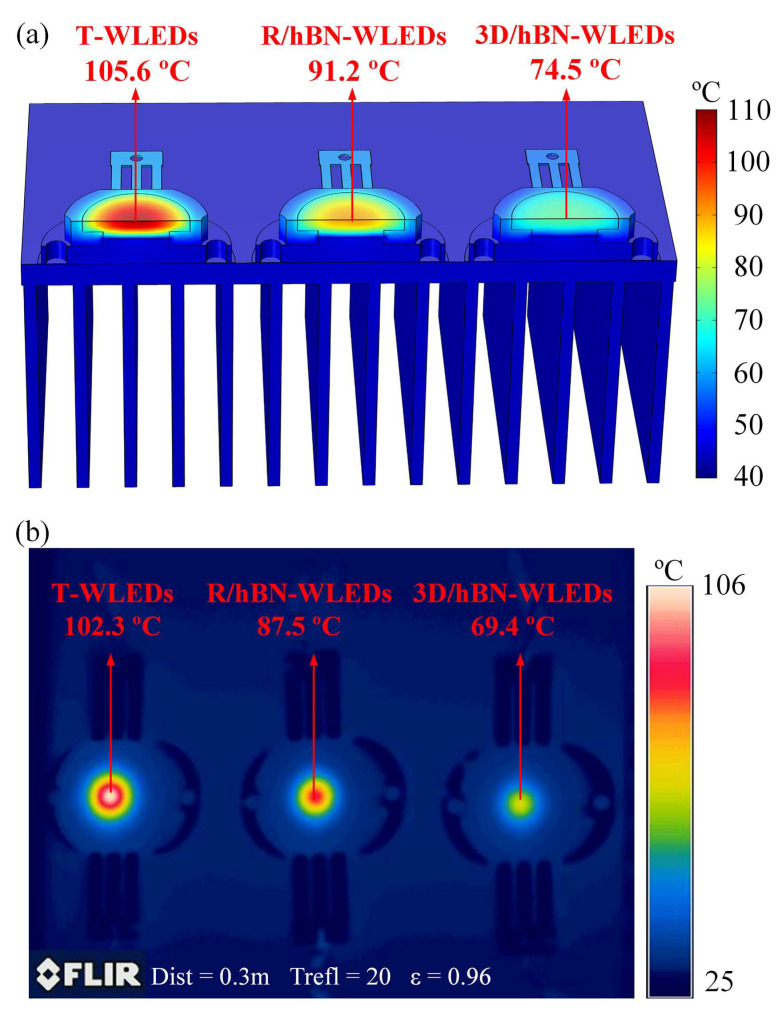
(**a**) Simulated and (**b**) measured temperature fields of these WLEDs under 500 mA illumination.

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
