# Peer review of "Enhancing Heat Dissipation of Photoluminescent Composite in White-Light-Emitting Diodes by 3D-Interconnected Thermal Conducting Pathways"

_micromachines, 2022, doi:10.3390/mi13081222_

Round 1
Reviewer 1 Report
Authors have developed a method to decrease the operational temperature of LEDs through increasing the thermal conductivity of the active polymeric composite. The idea is very appealing and important for increasing the performance and operational stability of the devices. There are few issues that should/can be improved though.
- In the 20th line of the abstract, you have to show hBN instead of BN to avoid confusion as you term it as 'hBN' throughout the abstract and manuscript;
-In the 30th line of the introduction, you don't need to show the abbreviation for visible light communication 'VLC' as you don't use it again in the manuscript;
-The explanation of the formation/emission of white light in the 34-39th lines of the introduction is too primitive and vague. You have to give more scientific and better detailed information on how the photons from different emitters are finally converted into one and totally different white light. In other words, you should give better details for white light generation process;
-Also, the information given in the 37-38th lines about "other blue light" that transmits through the composite without being absorbed seems at least less scientific to me. First, you have to avoid using words like "other" about the part of the blue light that is not absorbed and second, you have to explain why/how some parts of the blue light are absorbed and some parts aren't;
-The schematic shown in Fig. 1 does not illustrate the device structure clearly. Another thing is that you show an abbreviation 'PCB' which has no elaboration in the manuscript or in the figure caption.
-The word 'thermal-conducting' in the 58th line should be 'thermally conducting/conductive, I think;
-Do you mean 'transformed into hydrogel' in the 100th line or if it is 'transferred into hydrogel', then which/what kind of hydrogel did you use?
-The fabrication process given under 2.2. is inconsistent with Fig. 2. In the text you mention that you used 300 rpm and then 1500 rpm. However, it is 1300 rpm in the figure;
-You have to elaborate the formula shown in the 151st line and show what stands for what even if the equation is very conventional and simple;
-Another comment is about the work detailed under 3.1. How exactly did you optimize/control the volume fraction of air bubbles?
-You mention that the original shape of the 3D/hBN was retained. What about the size of the platelets?
-How was the simulation performed?
-And final comment, the results shown in Fig. 7 don't depict much difference. The difference between 84 and 85 and even 87 lm/W can be easily attributed to device to device variation rather than to improvement or optimization of the devices. And if I see it right, the traditional WLEDs with no hBN displayed better performance. Then what is point if the decrease of the operational temperature does not lead to an improvement in the performance? How many devices were measured to draw a conclusion?
Author Response
Reviewer 1:
Authors have developed a method to decrease the operational temperature of LEDs through increasing the thermal conductivity of the active polymeric composite. The idea is very appealing and important for increasing the performance and operational stability of the devices. There are few issues that should/can be improved though.
- In the 20th line of the abstract, you have to show hBN instead of BN to avoid confusion as you term it as 'hBN' throughout the abstract and manuscript;
Response: Sorry for that. We have corrected it in the revised manuscript.
-In the 30th line of the introduction, you don't need to show the abbreviation for visible light communication 'VLC' as you don't use it again in the manuscript;
Response: We have removed the abbreviation of VLC in the revised manuscript.
-The explanation of the formation/emission of white light in the 34-39th lines of the introduction is too primitive and vague. You have to give more scientific and better detailed information on how the photons from different emitters are finally converted into one and totally different white light. In other words, you should give better details for white light generation process;
Response: Thanks for the suggestion. We have made this description clearer in the revised manuscript.
-Also, the information given in the 37-38th lines about "other blue light" that transmits through the composite without being absorbed seems at least less scientific to me. First, you have to avoid using words like "other" about the part of the blue light that is not absorbed and second, you have to explain why/how some parts of the blue light are absorbed and some parts aren't;
Response: Thanks for the suggestion. We have made this introduction clearer in the revised manuscript.
-The schematic shown in Fig. 1 does not illustrate the device structure clearly. Another thing is that you show an abbreviation 'PCB' which has no elaboration in the manuscript or in the figure caption.
Response: We have revised Fig.1 to make the WLEDs structure clearer.
-The word 'thermal-conducting' in the 58th line should be 'thermally conducting/conductive, I think;
Response: Thanks for the suggestion. We have revised this description in the revised manuscript.
-Do you mean 'transformed into hydrogel' in the 100th line or if it is 'transferred into hydrogel', then which/what kind of hydrogel did you use?
Response: Sorry for the misleading description. We have corrected it as ‘transformed into hydrogel’.
-The fabrication process given under 2.2. is inconsistent with Fig. 2. In the text you mention that you used 300 rpm and then 1500 rpm. However, it is 1300 rpm in the figure;
Response: Sorry for the mistake. We have revised this description in Fig. 2.
-You have to elaborate the formula shown in the 151st line and show what stands for what even if the equation is very conventional and simple;
Response: Thanks for the suggestion. We have explained this formula in more detail.
-Another comment is about the work detailed under 3.1. How exactly did you optimize/control the volume fraction of air bubbles?
Response: The relative deviation of the volume fraction of air bubbles in different batches was controlled as less than 5%. We have added this description into the revised manuscript.
-You mention that the original shape of the 3D/hBN was retained. What about the size of the platelets?
Response: hBN platelets with average size of 45 μm and thickness of 1 μm were used in this work.
-How was the simulation performed?
Response: The simulation follows a standard finite element analysis approach which is conducted by a commercial software. Thus, we did not make extra description about this part.
-And final comment, the results shown in Fig. 7 don't depict much difference. The difference between 84 and 85 and even 87 lm/W can be easily attributed to device to device variation rather than to improvement or optimization of the devices. And if I see it right, the traditional WLEDs with no hBN displayed better performance. Then what is point if the decrease of the operational temperature does not lead to an improvement in the performance? How many devices were measured to draw a conclusion?
Response: It is right that T-WLEDs shows slightly higher luminous efficacy than the 3D/hBN-WLEDs. However, the 3D/hBN-WLEDs show evidently low working temperature than T-WLEDs. It is known that a lower working temperature is beneficial for the long-term stability of the WLEDs. Therefore, the 3D/hBN-WLEDs are advantageous in the aspect of lifetime.
9 samples were fabricated and tested for each kind of device.
Reviewer 2 Report
The study of Enhancing Heat Dissipation of Photoluminescent Composite in White Light-Emitting Diodes by 3D-Interconnected Thermal Conducting Pathways is reported, well conducted and give an overview of the material prepared and methodology designed. The conclusions are supported by the experimental evidence provided. However few suggestions are there which can improve the manuscript.
1. The details on the use of CNF for the fabrication of the composite not significantly highlighted. Is there any significant role of the CNF for the formation of 3D interconnected thermal conducting pathways? This may not be clearly emphasized in the abstract or introduction.
2. The novelty of the study indicating that no similar work has been carried out previously will be good to be highlighted.
3. The T-WLEDs, R/hBN-WLEDs, and 3D/hBN-WLEDs are mainly distinguished by the presence of hBN and the 3D-interconnected structure. What is the difference between R/hBN-WLEDs, and 3D/hBN-WLEDs in terms of the composite fraction? Such as the R/hBN-WLEDs also consist the CNF? It seems like CNF is included from fig 3 but proper specification will be better.
4. The effect of the hBN platelets to be interconnected and form 3D thermal conducting network undoubtedly may improve the thermal conductivity as compared to none and randomly distributed hBN without sacrificing the optical efficiency. This can be a promising finding for LED application. Nevertheless, the CNF has a very low thermal conductivity. Thus, can the 3D thermal conducting network form without the presence of CNF or using only the bubbles templating method? If no, the CNF play a significant role for the 3D network formation.
5. Line 70, … under an extremely hBN filling load of…please revise the sentence. Extremely low?
6. The authors mentioned Fig 4f shows the SEM images of the 3D/hBN-composite, in which the previous air voids of 3D/hBN-aerogel were completely filled by the QDs-silicone, however the author also indicated the presence of air void in figure 4f caption for the composite?
7. Some relevant references from Micromachines that could be helpful for the discussion, could be cited.
Author Response
Reviewer 2:
The study of Enhancing Heat Dissipation of Photoluminescent Composite in White Light-Emitting Diodes by 3D-Interconnected Thermal Conducting Pathways is reported, well conducted and give an overview of the material prepared and methodology designed. The conclusions are supported by the experimental evidence provided. However few suggestions are there which can improve the manuscript.
- The details on the use of CNF for the fabrication of the composite not significantly highlighted. Is there any significant role of the CNF for the formation of 3D interconnected thermal conducting pathways? This may not be clearly emphasized in the abstract or introduction.
Response: The CNF do have significant role in the formation of 3D interconnected pathways, because of its long molecular chains that can support and stabilize the as-formed 3D structures. We have emphasized this in the Introduction part of the revised manuscript.
- The novelty of the study indicating that no similar work has been carried out previously will be good to be highlighted.
Response: Thanks for the comment. We have made a clearer statement about the novelty of this work in the revised manuscript.
- The T-WLEDs, R/hBN-WLEDs, and 3D/hBN-WLEDs are mainly distinguished by the presence of hBN and the 3D-interconnected structure. What is the difference between R/hBN-WLEDs, and 3D/hBN-WLEDs in terms of the composite fraction? Such as the R/hBN-WLEDs also consist the CNF? It seems like CNF is included from fig 3 but proper specification will be better.
Response: The composite fractions between these WLEDs are slightly different. The R/hBN-composite contains silicone gel, phosphor, QDs, and hBN platelets, but no CNF, since the R/hBN-composite was fabricated by traditional mixing approach. We have made more explanations about this part in the revised manuscript.
- The effect of the hBN platelets to be interconnected and form 3D thermal conducting network undoubtedly may improve the thermal conductivity as compared to none and randomly distributed hBN without sacrificing the optical efficiency. This can be a promising finding for LED application. Nevertheless, the CNF has a very low thermal conductivity. Thus, can the 3D thermal conducting network form without the presence of CNF or using only the bubbles templating method? If no, the CNF play a significant role for the 3D network formation.
Response: It is true that the CNF features a relatively low thermal conductivity. However, the CNF is necessary since it function as a crosslinking and supporting agent to avoid the collapse of 3D hBN network. Besides, the CNF has neglectable influence on the overall thermal conductivity of the composite, because the thermal conductivity of the composite is mainly determined by the filling fraction as well as distribution of hBN platelets.
- Line 70, …under an extremely hBN filling load of…please revise the sentence. Extremely low?
Response: Sorry for the mistake. We have corrected it in the revised manuscript.
- The authors mentioned Fig 4f shows the SEM images of the 3D/hBN-composite, in which the previous air voids of 3D/hBN-aerogel were completely filled by the QDs-silicone, however the author also indicated the presence of air void in figure 4f caption for the composite?
Response: Sorry for the misleading statement. In Fig. 4f, the red dashed circles indicate the locations of previously existed air voids before the filling of QDs-silicone. We have revised this description in the revised manuscript.
- Some relevant references from Micromachines that could be helpful for the discussion, could be cited.
Response: Thanks for the suggestion. We have added several relevant references from Micromachines in the revised manuscript.
Reviewer 3 Report
In the manuscript entitled “Enhancing Heat Dissipation of Photoluminescent Composite in White Light-Emitting Diodes by 3D-Interconnected Thermal Conducting Pathways” where Puzhen Xia and coworkers reported the strategy to minimize the heat accumulation issue in photoluminescent composite, and thus improve the optical stability of WLEDs which is key for commercializing OLEDs. The manuscript is sufficient of originality and novelty, and the contents are represented appropriately based on the experimental results. However, there is minor revision with experimental evidence is required before publishing.
1. The author should elaborate on the novelty and originality of the work which is important in real-time application. Few lines should be added in the introduction part by considering the examples where WOLEDs are extensively used.
2. Lack of experimental data to strengthen their hypothesis. For example, the Bubbles templating method was used to construct three-dimensional (3D)-interconnected thermal conducting pathways inside the photoluminescent composite, under an extremely hBN filling load of 2.67 wt%. The thermal conductivity of the composite was efficiently enhanced by 100% from 0.158 to 0.318 W/(m∙K) due to continuous chain formation.
Figure 4 SEM images should be compared with T-composite, R/hBN-composite, and 3D/hBN composite to prove their concept.
Author Response
Reviewer 3:
In the manuscript entitled “Enhancing Heat Dissipation of Photoluminescent Composite in White Light-Emitting Diodes by 3D-Interconnected Thermal Conducting Pathways” where Puzhen Xia and coworkers reported the strategy to minimize the heat accumulation issue in photoluminescent composite, and thus improve the optical stability of WLEDs which is key for commercializing OLEDs. The manuscript is sufficient of originality and novelty, and the contents are represented appropriately based on the experimental results. However, there is minor revision with experimental evidence is required before publishing.
- The author should elaborate on the novelty and originality of the work which is important in real-time application. Few lines should be added in the introduction part by considering the examples where WOLEDs are extensively used.
Response: Thanks for the suggestion. We have made a clearer statement about the novelty of this work in the revised manuscript.
- Lack of experimental data to strengthen their hypothesis. For example, the Bubbles templating method was used to construct three-dimensional (3D)-interconnected thermal conducting pathways inside the photoluminescent composite, under an extremely hBN filling load of 2.67 wt%. The thermal conductivity of the composite was efficiently enhanced by 100% from 0.158 to 0.318 W/(m∙K) due to continuous chain formation.
Response: The thermal performance of the as-fabricated 3D pathways is mainly validated by the measurement of thermal conductivity, the temperature fields, and the microstructures. To make our hypothesis more convincible, we have added more microstructural characterizations in the revised manuscript.
- Figure 4 SEM images should be compared with T-composite, R/hBN-composite, and 3D/hBN composite to prove their concept.
Response: Thanks for the suggestion. We have added more SEM images to reveal the microstructural distinctions between these three composites.
Round 2
Reviewer 1 Report
Most of the concerns were addressed. However, I recommend to go through the manuscript once again. For instance, they added figure showing the fabrication process showing 1300 rpm and 1500 rpm speeds. But the text still shows 300 rpm and 1500 rpm.
Author Response
Reviewer 1:
Most of the concerns were addressed. However, I recommend to go through the manuscript once again. For instance, they added figure showing the fabrication process showing 1300 rpm and 1500 rpm speeds. But the text still shows 300 rpm and 1500 rpm.
Response: Thanks for the suggestion. We have unified the descriptions in the figure and text. One should be noted that, the '300 rpm' condition in the text is only for the pre-mixing process. For the foaming process, both the figure and text shows '1500 rpm'.